# Child Community Mental Health Services in Asia Pacific and Singapore’s REACH Model

**DOI:** 10.3390/brainsci7100126

**Published:** 2017-10-06

**Authors:** Choon Guan Lim, Hannah Loh, Vidhya Renjan, Jason Tan, Daniel Fung

**Affiliations:** Department of Child and Adolescent Psychiatry, Institute of Mental Health, Singapore 539747, Singapore; hannah_yl_loh@imh.com.sg (H.L.); vidhya_renjan@imh.com.sg (V.R.); Jason_ZX_TAN@imh.com.sg (J.T.); Daniel_Fung@imh.com.sg (D.F.)

**Keywords:** child psychiatry, community mental health, school-based mental health, mental health service

## Abstract

In recent decades, there have been concerted efforts to improve mental health services for youths alongside the challenges of rising healthcare costs and increasing demand for mental health needs. One important phenomenon is the shift from traditional clinic-based care to community-based mental health services to improve accessibility to services and provide patient-centred care. In this article, we discuss the child and adolescent community mental health efforts within the Asia-Pacific region. We also discuss Singapore’s community and school-based mental health service, known as the Response, Early Intervention and Assessment in Community Mental Health (REACH). This article discusses how REACH has evolved over the years in response to the changing needs of youths in Singapore. Finally, we discuss the current challenges and future directions for youth mental health care.

## 1. Burden of Mental Health in Young People

### 1.1. Overview

It is reported that 1 in 10 children experience mental health problems but less than a third of them are likely to seek help [1]. Furthermore, mental health disorders in youths are often not detected till much later in life [2]. Globally, mental disorders account for a significant portion of disease burden in youths, with poor mental health impacting the young person’s physical health and development by affecting their academic and vocational achievements, social relationships, and exposing them to stigma and discrimination [1,2,3]. Poor youth mental health is also significantly associated with maladaptive substance use, violence and increased mortality rates due to suicide, self-harm, and accidental harm which further increases health-care costs [1,2].

### 1.2. Greater Push towards Improving Mental Health Support for Children and Adolescents

Historically, a larger proportion of mental health services and resources have been channeled towards treating older and chronic populations. Treating youths required specialized child-friendly and youth-oriented therapeutic approaches with few psychiatrists and professionals choosing to specialize in this area, particularly in the Asia-Pacific region. Diagnosing youths can be challenging due to multiple psycho-social influences, and often requires a multi-disciplinary, inter-agency approach to treatment and care [2].

However, over the past four decades, efforts to improve the mental health treatment and services for children and adolescents have grown considerably [4]. Increasingly, governments and global healthcare organizations, such as the World Health Organization (WHO), recognize the burden of mental disorders in youths and have moved towards prevention, earlier identification and intervention, and improving mental health resilience amongst youths [5,6]. For example, in 2005 the WHO promulgated its Mental Health Policy and Service Guidance Package to provide specific guidelines to healthcare makers and governments on policy development and service delivery of child and adolescent mental health services [7].

### 1.3. Singapore

Singapore is a small island-state located within Southeast Asia. Its population is multi-ethnic, with Chinese making the majority at 74%, followed by Malays (13%), Indians (9%), and other ethnic minorities (3%). As of 2016, Singapore had approximately 835,900 children and adolescents below the age of 20 years [8]. In the only community-based prevalence study involving 2139 school-going children aged 6 to 12 years, the prevalence of emotional and behavioral problems was found to be comparable to studies in the West at 12.5% [9]. The same study also found the prevalence of internalizing disorders to be more than twice that of externalizing disorders which was in contrast to studies in the West. In a 2010 study on disease burden amongst Singapore residents, mental disorders accounted for the largest portion, about one-third of disability-adjusted life years (DALYs), in children aged below 15 years. Specifically, autism spectrum disorder, attention-deficit/hyperactivity disorder, and anxiety and depression were amongst the top 10 causes of disability burden amongst children between the ages of 0 and 14 years. Amongst young people between the ages of 15 and 34 years, mental disorders—mainly schizophrenia, and anxiety and depression—accounted for a significantly greater burden, approximately 19% DALYs, within this age group [10]. Suicide rate is one of the surrogate indicators to measure mental well-being of a society. Among young people in Singapore, the suicide rate was 5.7 per 100,000 and closely associated with psychosocial stressors, including academic stress [11,12]. An analysis of global suicide rates in 81 countries, between 1990 and 2009, showed a declining trend in average suicide rates of youths aged between 15 and 19 years. Globally, the average suicide rate for males decreased from 10.32 to 9.50 (per 100,000) and the suicide rate for females remained steady at 4.41 to 4.19 (per 100,000). In Europe, the average suicide rate showed a significant decline for both genders, decreasing from 13.13 to 10.93 (per 100,000) in males and from 3.88 to 3.34 (per 100,000) in females. In the United States, there was a significant decrease in suicide rates for males (16.13 to 11.81 per 100,000) and females (3.31 to 2.82 per 100,000). In Australia, there was a significant decline in suicides for males 15 to 19 years (16.79 to 11.10 per 100,000), while rates remained stable for females (4.12 to 4.17 per 100,000). However, the authors note that limited data exists for African and Asian countries, and further examination of suicide rates in these regions is required to aid in cross-cultural comparisons [13].

Demand for services at the outpatient child and adolescent psychiatric clinic (Child Guidance Clinic, or CGC) of the Institute of Mental Health (IMH), Singapore’s only tertiary-care psychiatric hospital, has steadily increased over the years. The number of new cases seen has risen six-fold from approximately 500 new cases annually in the 1980s to almost 3000 annually by 2010 [14]. This increase is concerning and while it may be due to the clinic’s outreach effort to the schools and greater awareness of mental disorders in the community rather than an actual increase in the incidence of mental disorders, it exerts a strain on the clinic’s resources. Additionally, to improve clinical outcome, there is a need to focus on prevention and providing care further upstream. The current clinic-based model does not seem ideal to fulfill these roles.

## 2. Global Trends in Community Mental Health Efforts

Past models of care depended on mental health professionals for direct service delivery. However, the shortage of trained mental healthcare professionals make provision of future direct services in outpatient and inpatient settings unsustainable due to rapid growing demands and costs. There is also pressure on mental health services to provide appropriate and timely services that are sustainable and cost-effective [15]. With the push to deinstitutionalize mental healthcare and move towards a tiered approach in mental healthcare provision, there is a shift towards establishing community-based mental health services [16].

The WHO’s Comprehensive Mental Health Action Plan 2013–2020, for example, calls for member states to “provide comprehensive, integrated and responsive mental health and social care services in community-based settings” to improve access of care and service quality [17]. Other benefits of community-based care include early access to intervention, increased treatment adherence, human rights protection, prevention of stigma, reduced hospital admissions, and fewer deaths by suicide [18].

## 3. Child and Adolescent Community Mental Health Efforts across the Asia-Pacific Region

Community mental healthcare is still in a nascent stage of development in the Asia-Pacific region, with governments and service providers working collaboratively together only in the last decade to increase funding, and establish policies and guidelines on community mental healthcare. Such developments include the establishment of the Asia-Pacific Community Mental Health Development (APCMHD) project in 2005 by 14 member nations to promote information exchange on best practices within the region [19].

### 3.1. Increased Accessibility to Community Mental Health Teams and Centers

A key feature of many community mental health models is the establishment of mental health teams and centers within the community to increase accessibility of services. Community mental health teams are usually multi-disciplinary in nature comprising psychiatrists, psychologists, occupational therapists, social workers, nurses, and other allied health professionals, who are situated in community mental health centers located conveniently in districts and bureaus across the country.

For instance, in Hong Kong, the Social Welfare Department introduced the Integrated Community Center for Mental Wellness (ICCMW) initiative in 2010, a one-stop, integrated, district-based center where people with mental health problems can access mental health services at designated centers across different districts [20].

Increasingly, community mental health teams have focused specifically on servicing children and adolescents. As of 2004, 24 community mental health centers in South Korea engaged in school mental health services to neighboring schools [21]. In Taiwan and Singapore, specialized early intervention centers have been established since the mid-2000s to provide assessment and intervention services to children with developmental delays and their families [22].

### 3.2. Increased Patient-Centered Innovation

A shift towards community care also signals a shift in attitude towards organizing care around patients’ needs. Increasingly, governments and healthcare providers are designing services that are integrated, comprehensive, flexible, and responsive to overcome challenges that patients encounter in accessing mental health services, including stigma, proximity to services, and lack of continuity of care. In Taiwan for example, the “Taipei Model” developed by the Taipei City Psychiatric Center (TCPC) seeks to integrate community mental healthcare services within the primary care and general healthcare system. Under this model, public health workers from 12 district health centers provide follow-up visits to patients who were recently discharged from the TCPC to ensure that there is continuity of care to patients [19].

To make mental healthcare services accessible to youths living in rural parts of Australia, the “headspace” program was established in 2006 to minimize stigma and aid in the promotion and early intervention of mental health problems for youth aged 12 to 25 years [23]. Additionally, “headspace” was designed to integrate mental and physical health services to provide a one-stop stigma-free service for youths who often presented with a comorbid of physical and mental health problems. In an evaluation study of “headspace”, both clinicians and youths reported that the co-location of medical and counselling services within the service encouraged help-seeking behavior, while youths reported increased likelihood of treatment adherence [24].

Moreover, mental healthcare providers in Australia have innovated services to meet the needs of the population. For instance, “ReachOut”, established in 1998, is Australia’s leading online mental health service for youths which aims to provide youths with easy access to 24/7 anonymous help by employing online technology [25].

### 3.3. Increased Government Funding and Support

Across Asia, governments have also increased funding earmarked for community mental health efforts, and pushed for a more comprehensive system of care. Recently in Singapore, an additional funding of S$160 million has been ring-fenced to increase community mental health services as part of a five-year Community Mental Health Masterplan from 2017 to 2021. Governments in the region have also pushed for preventative care and early intervention as part of community-based mental healthcare, targeting children and adolescents. For example, the South Korean 2005 Mental Health Services guidelines recognizes that critical aspects of child and adolescent mental health services include the prevention of mental health problems, early detection of such problems, and easier access to suitable treatments [21].

## 4. Singapore’s Response

From a Singapore perspective, many mental disorders, including anxiety and depressive disorders, have onset in the childhood and adolescent years but remain untreated [26]. This emphasizes the need for early identification and intervention. Under the traditional model of care, a number of challenges with healthcare service delivery for youths were highlighted, such as long wait times, lack of appropriate and specialized care, fragmentation of service delivery within public health and social agencies, and duplication of services [27]. Stigma also notably plays a significant role for many Singaporean families, which may lead some to seek practitioners of traditional medicine or spiritual healers, consequently delaying access to appropriate and timely help [26,28]. Voluntary welfare organizations and initiatives, such as the Silver Ribbon Project and Singapore Association of Mental Health (SAMH), have been active in addressing stigma through public education, awareness programs and encouraging seeking early treatment. Furthermore, these programs portray individuals with mental health issues in a positive light and focus on building self-esteem and psychological health aspects. In addition, public education programs are also conducted in schools to promote mental health literacy. For youths between 16 and 30 years, the Community Health Assessment Team (CHAT), located at SCAPE Youth Park in the main city district, provides a one-stop center (CHAT Hub) for mental health assistance and resources [29].

In 2007, the National Mental Health Blueprint was developed to encourage primary prevention and monitoring of mental health of the population, improving service delivery and quality of psychiatric services, developing the mental health workforce, and promoting mental health research. Part of the vision under the 2007–2012 National Mental Health Blueprint included the right-siting and deinstitutionalization of care, leading to the inception of community-based services.

## 5. The REACH Model of Care

### 5.1. The Inception and Initial Years of REACH

In 2007, Response, Early Intervention and Assessment in Community Mental Health (REACH) was developed to support school-going students with mental health problems. Since 2003, compulsory education was enforced in Singapore and today, the compulsory school age is 6 to 15 years. The three main objectives of REACH are:improve mental health of youth via early assessment and intervention;build the capacity of schools and community partners to detect and manage mental health problems through support and training;build a community mental health support network for children and adolescents in the community, consisting of schools, general practitioners (GPs), and voluntary welfare organizations (VWOs).

The REACH model (see Figure 1) operates on regional hospital systems to create regional networks of mental healthcare and social services for seamless continuum of care for children and their families. The conception of REACH creates responsive services based on five criteria of quality care that is effective, accessible, timely, affordable, and safe as the central support mechanisms.

REACH consists of four teams that service the north, south, east, and west geographical zones of Singapore, corresponding to the school zones [28]. Each team is comprised of a mobile multi-disciplinary team of medical doctors, psychologists, medical social workers, occupational therapists, and psychiatric nurses. REACH provides its services to schools primarily, and will provide assessment and appropriate intervention for students referred by the school counselor for mental health concerns. They also partner and engage VWOs (non-profit organizations that provide welfare services and/or services that aid the community) in providing services to these students and their families. By tapping on the provision of allied health services in schools, delivery of care via schools is envisioned to aid and increase mental wellness among school students. The REACH program is gradually phased to allow refinement of the program at each step.

The involvement of schools and community agencies reflects the global shift to a multi-sectoral and multi-agency tiered approach of care. Within school support services, teachers with basic counseling and behavioral management skills provide first-level intervention to students with socio-emotional and behavioral problems. At the next level, allied educators (mainstream schools) and allied health professionals (special schools) provide care to students who required specialized attention. For those who required intensive intervention or attention may be referred to specialists at the Ministry or external agencies such as hospital-based clinics or VWOs. Traditional clinic-based care for youths with mental health problems almost always involves collaborative care with schools and/or social agencies. Through the community-based care provided by REACH, the physical barrier in care coordination was bridged. Existing support networks further strengthened REACH teams to be mobile and provide assessment and intervention within schools or at home, thus, increasing accessibility of services and understanding of the child’s problems within their naturalistic environment. Furthermore, the accessibility of REACH teams helps to reduce default rates of appointment, thereby increasing adherence to treatment and time to seek help [27].

The consultation helpline plays an integral role in REACH’s service delivery. First, it increases timely accessibility to care. Secondly, cases that are referred via the helpline are tabled for discussion and triaged according to priority. School counsellors first gather relevant preliminary information regarding the child’s mental health prior to contacting the REACH consultation helpline. However, there are limitations related to the varying clinical knowledge and understanding of school counsellors. Nevertheless, school counsellors and REACH teams work in collaboration to ensure that critical referral information is gathered. It should also be noted there is a designated REACH member assigned to each school who provides consultation to school counsellors when required. The referral is then tabled for discussion with the rest of the REACH multidisciplinary team to determine its appropriateness for referral for a more comprehensive mental health assessment. Cases that are deemed to have insufficient evidence for a mental health problem are provided recommendations for alternative support. This ensures an effective utilization of the team’s resources. Apart from providing care, REACH also focuses on building the clinical capacity of school counsellors, VWOs, and community partners, by providing specialist training in early detection and management of common mental health problems seen among children and adolescents. Community partners of REACH are also invited to trainings by overseas mental health experts and monthly inter-agency case conferences to increase their clinical knowledge and aptitude in detection and management of mental health needs [27]. The engagement and collaboration of the network of agencies ensures that fragmentation of care and larger systemic issues are addressed. Table 1 shows the breakdown of diagnoses seen by REACH in 2015. Attention-Deficit Hyperactivity Disorder (ADHD) was the most commonly seen mental health disorder. Emotional disorders, including anxiety and mood disorders, were the second most commonly seen disorders by REACH teams.

In a review of the effectiveness of the REACH program in terms of outcomes and effectiveness between 2007 and February 2015, 4184 students were referred to REACH by school counsellors. Out of these students, about 955 students were referred to tertiary specialist care. Clinical outcomes are measured by counsellor and teacher ratings on the Clinical Global Impression (CGI) scale [30] and Strength and Difficulties Questionnaire (SDQ; [31]), respectively. These measures are completed at intake assessment and six months post-assessment, regardless of whether the child receives intervention. These outcomes are presented in Table 2. In general, improvements in conduct problems, emotional problems, hyperactive behaviors and peer problems, and prosocial behavior on the SDQ and CGI severity of illness significantly improved after six months. Furthermore, findings show that community-based care compared to hospital-based care was more cost-effective, at a negative incremental cost-effectiveness ratio of S$18,308 per quality-adjusted life year (QALY) gained and maintained cost-effectiveness over the 95% confidence interval of QALY estimates [32,33].

Anecdotally, REACH services appear to be well-received by schools, as it is based in the community setting and is less stigmatizing in providing early assessment and intervention for youths. School leaders and counsellors have also given positive feedback on REACH teams’ responsiveness to their schools’ needs. In addition, schools have commended REACH team members’ professional knowledge, communication skills, and provision of recommendations and care management plans to address the needs of each child. Conversely, for school staff who may not be aware of REACH services, REACH continues to generate awareness of its services through regular mental health literacy trainings to teachers in schools. Regular dialogues and reviews are also conducted inter-ministry, between REACH representatives (falling under the purview of the Ministry of Health) and the Ministry of Education (MOE), to address the changing needs of students and schools.

### 5.2. The Evolution of REACH

Since its inception, REACH has evolved to keep up with the changing needs and demands. There have been notable service developments to increase collaboration with community agencies and caregivers of the students seen by REACH. These included greater engagement with the MOE to streamline referral processes and improve inter-professional liaison and joint management of cases, partnership with specialized intervention centers to provide evidence-based intervention programs for children with co-morbid conditions such as dyslexia and conduct problems. Through collaboration, REACH also provides training workshops to equip school counselors and community partners with technical knowledge about specific mental illness, and intervention techniques such as motivational interviewing, mindfulness, and sensory-based strategies.

In 2016, REACH started a triage service for outpatient referrals made by regional polyclinics to CGC so as to arrange suitable cases for school-based assessment. Out of the 820 referrals from polyclinics, 279 (34%) were eventually assessed by REACH. Consequently, first visit wait-times were reduced at CGC. Re-directing appropriate referrals to REACH also aids in increasing catchment areas for responsive service provision.

## 6. Challenges and Future Considerations

On a global scale, major treatment gaps in addressing mental health problems exist, including a lack of manpower and clinical capacity of the mental healthcare workforce, varying political support of mental healthcare infrastructure and implementation of policies, and evaluation of novel and accessible models of care [34].

In Singapore, the REACH model of care has progressed to provide effective, timely, and accessible mental health service to children and adolescents. Despite the positive impact of the program, future directions include improving the processes in right-siting of referrals, continued capacity building of the REACH team and our stakeholders, and expansion of our collaboration with more school partners.

In addition to triaging outpatient referrals, REACH has also been looking to offer more group-based therapies to treat students presenting with mild mental health needs to more efficiently meet the increased demand of services. Also, REACH is looking towards leveraging information technology systems to enhance all areas of service delivery. This includes using data analytic techniques to analyze trends in case referrals, and possibly tapping into technology-based assessment and therapy platforms. While initially conceptualized to provide services to children with mild mental health needs, the REACH program has now come to service children with more complex needs who prefer community care as opposed to secondary and tertiary psychiatric care. To meet the challenge of assessing and treating students with more complex difficulties, REACH team members would have to continually build competence through regular case discussions, supervision, and training. Besides having a background of providing cognitive-behavioral therapy, REACH psychologists are building competencies in third-wave therapies including mindfulness and acceptance and commitment therapy, whilst REACH occupational therapists build on sensory integration techniques, and occupational engagement. Given the larger systemic issues seen in children, REACH team members also receive training on evidence-based parenting programs such as the “Incredible Years” [35] and “Triple P” [36] programs. REACH also recognizes that fellow school and community partners have differing levels of competence which poses a challenge of catering to their varying training needs. Newer school and community partners may require more extensive training in basic mental health support strategies and require greater support from REACH teams.

As part of our commitment to reach out and make mental healthcare more accessible, future plans include expanding collaborations to a larger age-range by including preschools, and institutes of higher learning (e.g., polytechnics and universities). Concurrently, REACH continues to strengthen current partnerships with schools by promoting the development of whole-school approaches to support positive mental health and resilience in children by enhancing mental health literacy in the curriculum, and fostering a resilient culture that supports positive emotional well-being.

Another important aspect is the exchanging of our experience in providing school-based mental health care with our south-east Asian and Asian colleagues. While we may share some similar culture, educational, and health systems do vary from country to country. Over the years, we have shared our experiences in several regional conferences in China, Japan, Korea, Malaysia, and Indonesia. We will continue to maintain our professional networks so that we can continue to be updated on regional developments in youth mental health care services, especially in the context of community mental health services. For example, in the local residency program for child and adolescent psychiatry, overseas clinical fellowships are encouraged to enrich the experience of the next generation of psychiatrists.

## Figures and Tables

**Figure 1 brainsci-07-00126-f001:**
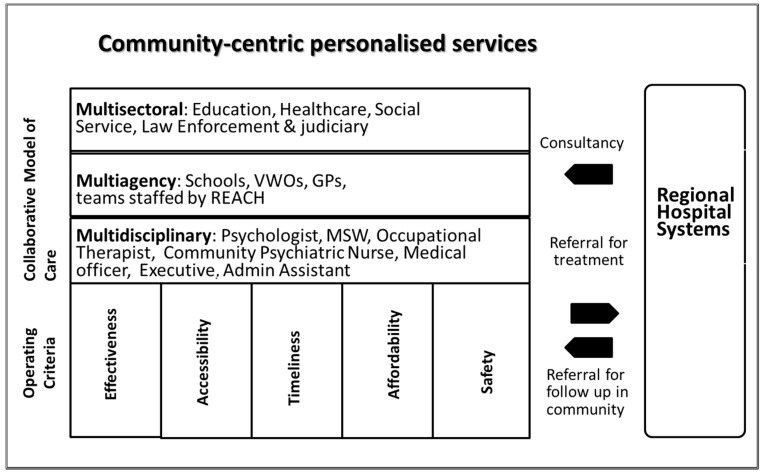
The REACH model of care. Note: VWOs = Voluntary Welfare Organizations; GPs = General Practitioners; MSW = Medical Social Worker.

**Table 1 brainsci-07-00126-t001:** Breakdown by diagnosis for the cases seen in 2015.

Diagnosis	No. (%) of Cases
Attention-deficit hyperactivity disorder	247 (35.7)
Emotional disorder	159 (23.0)
No mental illness	133 (19.2)
Adjustment disorder	69 (10.0)
Developmental disorder	36 (5.2)
Conduct disorder	29 (4.2)
Others	18 (2.6)

Note: Emotional disorder includes anxiety and mood disorders; Adjustment disorder includes all stress-related disorders including situational reaction and adjustment disorder); Developmental disorder includes learning/developmental disorders and autism spectrum disorders; Conduct disorder includes cconduct disorder, oppositional defiant disorder, and mixed emotional and conduct disorder.

**Table 2 brainsci-07-00126-t002:** Means and standard deviations of clinical outcomes at pre and post (six-month) assessment.

	Pre-Assessment		Post Six-Month		
	Mean	SD	Mean	SD	*t*-test
CGI	3.14	0.11	2.42	0.06	17.88 **
SDQ-Emotional Problem	2.76	0.33	2.13	0.33	3.51 **
SDQ-Conduct	3.17	0.56	2.61	0.70	7.34 **
SDQ-Hyperactivity	6.71	0.46	6.01	0.41	12.18 **
SDQ-Peer problem	3.93	0.22	3.45	0.31	10.01 **
SDQ-Prosocial	4.05	0.30	4.47	0.70	−2.27 *

*Note*: CGI = Clinical Global Impression scale. SDQ = Strength & Difficulties Questionnaire. * *p* = 0.05, ** *p* < 0.01. Higher scores on the SDQ-prosocial behavior scale reflect desired behaviors.

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
