# Peer review of "Child Community Mental Health Services in Asia Pacific and Singapore’s REACH Model"

_brainsci, 2017, doi:10.3390/brainsci7100126_

Round 1

Reviewer 1 Report

As authors described, this article would be valuable to understand the natural shift of mental health service paradigm from clinic-based care to community health service.

I recommend to add (1) the legend about MSW in figure 1, (2) references about CGI and SDQ in main text, and (3) references or footnotes about 'Incredible Years' and "Triple P' programmes.

Also, it needs more comments or descriptions in [6. Challenges and Future Consideration] for Asia Pacific region beyond Singapore model.

Author Response

Dear Sir/Mdm

RE: Manuscript ID: brainsci-218466 - Minor Revisions

We would like to thank the editor’s efforts and helpful comments from the reviewers. We will address the comments in chronological order and highlight the amendments which have been made in the manuscript.

Peer Review Report 1

1.                  I recommend to add (1) the legend about MSW in figure 1, (2) references about CGI and SDQ in main text, and (3) references or footnotes about 'Incredible Years' and "Triple P' programmes.

Response: We have added the legend in Figure 1 as recommended. As suggested we have included the relevant references for CGI and SDQ on page 6 of the manuscript, as references numbered 30 and 31 respectively. Similarly we have also included the relevant references for 'Incredible Years' and "Triple P' programmes on page 8 of the manuscript, as references numbered 35 and 36 respectively.

2.                  Also, it needs more comments or descriptions in [6. Challenges and Future Consideration] for Asia Pacific region beyond Singapore model.

Response: We have, as suggested, included a paragraph at the end of this section, about networking and cross-learning with our colleagues from the Asia Pacific region.

Thank you.

Yours Sincerely,

 Dr Lim Choon Guan

Reviewer 2 Report

A much-needed description of a much-needed service innovation.  Worth more elaboration: 1) how does suic rate in SG (given) compare to other countries?; 2) more about extent of and ways of dealing w MH stigma in SG communities; 3) despite six-fold incr in referrals to CGC, overall rate still seems low; 4) re outcomes:  in addn to CGI and SDQ, any data re other outcomes - suicide rate, school drop out rate, juvenile offending rate?  5)  Readers would be int'd to know how REACH has been received in the schools - all teachers enthusiastic?  What about those who are not? Also minor text fixes needed at line 140 and line 226-228.

 In this way of spreading MH services more broadly, bringing school-based counselors into important roles, the question of triage - how to identify those children and youth needing a more highly trained specialist - always comes us.  Many readers of the article would be interested to know how that triage function has been structured and of how it's gone.

Author Response

Dear Sir/Mdm

RE: Manuscript ID: brainsci-218466 - Minor Revisions

We would like to thank the editor’s efforts and helpful comments from the reviewers. We will address the comments in chronological order and highlight the amendments which have been made in the manuscript.

1.                  How does suicide rate in SG (given) compare to other countries?

Response: We have added global and regional trends in suicide rates from 81 countries. Please refer to lines 64-74.

2.                  More about extent of and ways of dealing with mental health stigma in Singapore communities.

Response: We have included information about stigma in mental illness in lines 162-169.

3.                  Despite six-fold increase in referrals to CGC, overall rate still seems low.

Response: On a personal note to the reviewer, this emphasizes the importance of our school-based approach which involves educating school counselors and teachers to identify symptoms of mental illnesses and improve accessibility to care. Though the largest, Child Guidance Clinicis not the only centre in Singapore providing child and adolescent mental psychiatry services. There are also a few restructured hospitals providing such care, as well as the paediatric medical services. Unfortunately, we do not have the number of youths attending such services. 

4.                  Re outcomes:  in addition to CGI and SDQ, any data re other outcomes - suicide rate, school drop out rate, juvenile offending rate? 

Response: At present, we do not capture these data but agree that they will be useful to be considered as we review the outcome measures for the REACH program in the future.

5.         Readers would be int'd to know how REACH has been received in the schools - all teachers enthusiastic?  What about those who are not? 

Response: Anecdotally, REACH services appear to be well-received in general by schools as it is based in a community setting and less stigmatizing with regard to providing early assessment and intervention for youth. With those who are not, we provide ongoing awareness of REACH services and its benefits and mental health literacy trainings to teachers in schools. We included this at lines 263 – 270.

6.         Re: Also minor text fixes needed at line 140 and line 226-228.

Response: Minor text revisions have been made at lines 149-152 and also at lines 248-251.

7.         In this way of spreading MH services more broadly, bringing school-based counselors into important roles, the question of triage - how to identify those children and youth needing a more highly trained specialist - always comes us.  Many readers of the article would be interested to know how that triage function has been structured and of how it's gone. 

Response: The first line assessment is done by the school counsellors to gather relevant information prior to contacting the REACH consultation helpline to make a referral. However, we recognize that there are limitations related to the varying clinical knowledge and understanding of counsellors. Nevertheless,  school counsellors and REACH team work in collaboration to ensure that critical information is gathered. To build the capacity of our school counselors, we provide training workshops, invite them to attend regular inter-agency case conferences. There are also designated REACH personnel who can be the resource person for each counselor, REACH can assess if needed and decide. 

We have expanded on this section (kindly refer to lines 220-227 onwards in the amended manuscript).

Thank you.

Yours Sincerely,

 Dr Lim Choon Guan